# Dysfunction of EAAT3 Aggravates LPS-Induced Post-Operative Cognitive Dysfunction

**DOI:** 10.3390/membranes12030317

**Published:** 2022-03-11

**Authors:** Xiao-Yan Wang, Wen-Gang Liu, Ai-Sheng Hou, Yu-Xiang Song, Yu-Long Ma, Xiao-Dong Wu, Jiang-Bei Cao, Wei-Dong Mi

**Affiliations:** 1Chinese PLA Medical School, Beijing 100853, China; wxyansmile2022@163.com (X.-Y.W.); cctvzjkx@163.com (W.-G.L.); 2Department of Anesthesiology, The Fourth Medical Center of Chinese PLA General Hospital, Beijing 100037, China; 3Department of Anesthesiology, The First Medical Center of Chinese PLA General Hospital, Beijing 100853, China; houaisheng301@163.com (A.-S.H.); kent_song@126.com (Y.-X.S.); yulongma123@163.com (Y.-L.M.); wuxd301@163.com (X.-D.W.)

**Keywords:** excitatory amino acid transporter 3 (EAAT3), post-operative cognitive dysfunction (POCD), lipopolysaccharide (LPS), riluzole, aging

## Abstract

Numerous results have revealed an association between inhibited function of excitatory amino acid transporter 3 (EAAT3) and several neurodegenerative diseases. This was also corroborated by our previous studies which showed that the EAAT3 function was intimately linked to learning and memory. With this premise, we examined the role of EAAT3 in post-operative cognitive dysfunction (POCD) and explored the potential benefit of riluzole in countering POCD in the present study. We first established a recombinant adeno-associated-viral (rAAV)-mediated shRNA to knockdown *SLC1A1*/EAAT3 expression in the hippocampus of adult male mice. The mice then received an intracerebroventricular microinjection of 2 μg lipopolysaccharide (LPS) to construct the POCD model. In addition, for old male mice, 4 mg/kg of riluzole was intraperitoneally injected for three consecutive days, with the last injection administered 2 h before the LPS microinjection. Cognitive function was assessed using the Morris water maze 24 h following the LPS microinjection. Animal behavioral tests, as well as pathological and biochemical assays, were performed to clarify the role of EAAT3 function in POCD and evaluate the effect of activating the EAAT3 function by riluzole. In the present study, we established a mouse model with hippocampal *SLC1A1*/EAAT3 knockdown and found that hippocampal *SLC1A1*/EAAT3 knockdown aggravated LPS-induced learning and memory deficits in adult male mice. Meanwhile, LPS significantly inhibited the expression of EAAT3 membrane protein and the phosphorylation level of GluA1 protein in the hippocampus of adult male mice. Moreover, riluzole pretreatment significantly increased the expression of hippocampal EAAT3 membrane protein and also ameliorated LPS-induced cognitive impairment in elderly male mice. Taken together, our results demonstrated that the dysfunction of EAAT3 is an important risk factor for POCD susceptibility and therefore, it may become a promising target for POCD treatment.

## 1. Introduction

Post-operative cognitive dysfunction (POCD) is a postoperative complication of the central nervous system (CNS) that tends to occur in elderly patients who have received anesthesia and surgery [1,2]. Characterized by anxiety, personality changes, and memory impairment, POCD can significantly affect the postoperative recovery of patients, and increase postoperative morbidity and mortality as well as hospital expenses [3]. Though there are abundant studies about POCD, the main reason for the susceptibility among elderly patients has not yet been identified.

Excitatory amino acid transporter 3 (EAAT3), also known as excitatory amino acid carrier 1 (EAAC1), is a member of the family of Na^+^-dependent excitatory amino acid transporters (EAATs) that maintain extracellular Glu homeostasis in CNS [4,5]. EAAT3, encoded by the *SLC1A1* gene, ubiquitously exists in the brain and is enriched in the neurons of hippocampus and cortex [6,7]. Previous studies have found out that EAAT3 dysfunction may lead to neuropsychiatric diseases, such as Alzheimer’s disease (AD), Parkinson’s disease (PD), amyotrophic lateral sclerosis (ALS), Huntington’s disease, epilepsy, and schizophrenia [8,9,10,11,12,13]. Evidence has also shown that *SLC1A1*/EAAT3 overexpression could cause pathologic effects on dopaminergic neurotransmission and increase the risk of obsessive-compulsive disorder (OCD) [14,15]. Our previous study has found out that EAAT3 could regulate the transport of GluA1 to plasma membrane, which is considered to be critical to memory formation. However, the relationship between EAAT3 dysfunction and POCD has not yet been clarified.

Several preclinical and human studies have revealed that neuro-inflammation plays an important role in the progression of POCD [2,16]. Lipopolysaccharide (LPS) is a major bacterial TLR4 ligand that can activate innate immunity and induce inflammatory responses [17]. Neuroinflammation induced by LPS are well known experimental paradigms to explore the mechanisms of POCD. Accumulating studies have shown that intracerebroventricular administration of LPS could cause cognitive deficits by the upregulation of pro-inflammatory cytokines and subsequent alteration of redox homeostasis, which can be effectively used to establish POCD models in mice [18,19,20]. Riluzole (2-amino-6-(trifluoromethoxy) benzothiazole) is a glutamate transporter activator [21] approved by the FDA for ALS therapy. So, we wonder whether riluzole can alleviate LPS-induced POCD in old mice and what its underlying molecular mechanism is.

In this study, a recombinant adeno-associated viral (rAAV) -mediated shRNA was constructed to knockdown *SLC1A1*/EAAT3 in adult mice, and an LPS-induced cognitive impairment model (POCD animal model) was established to identify the function of EAAT3 during the development of POCD and its underlying mechanisms. Besides, the study also explored riluzole’s role in improving LPS-induced cognitive impairment in old mice. The results of this study may provide new insights into the prevention and treatment of POCD and its underlying mechanisms.

## 2. Materials and Methods

### 2.1. Animals

Adult (3 months, weighing 22–25 g) and old (21 months, weighing 28–36 g) male C57BL/6 mice were purchased from Beijing SPF Animal Technology Company (Beijing, China. Permit Number: SCXK1026-0002). The animals (2–4 mice per cage) were housed in a temperature- and humidity-controlled room with a standard 12–12 light/dark cycle as well as food and water. All animals were allowed to acclimatize to the environment for one week before the experiments and were fixed cage-mates throughout the acclimation and testing periods. All experimental procedures were approved by the Animal Care Committee of the Chinese People’s Liberation Army General Hospital (Beijing, China). All animal experiments were carried out in accordance with the current laws of China and the National Institutes of Health Guide for the Care and Use of Laboratory Animals.

### 2.2. Drugs

The artificial cerebrospinal fluid (ACSF) vehicle contained 140 mM NaCl, 3.0 mM KCl, 2.5 mM CaCl_2_, 1.0 mM MgCl_2_, and 1.2 mM Na_2_HPO_4_. LPS (Sigma, St. Louis, MO, USA) was dissolved in ACSF (1000 ng/mL).

Riluzole (Cayman Chemical Company, Ann Arbor, MI, USA) was first dissolved in dimethylsulphoxide (DMSO), (Sinopharm Chemical Reagent Co., Beijing, China) to 100 mmol/L (27 mg/mL) and then diluted in saline to 0.4 mg/mL with gentle warming. Furthermore, 4 mg/kg riluzole was injected intraperitoneally 2 h before LPS microinjection. A similar concentration of DMSO (1.6%) was used as solvent control.

### 2.3. Experiment Design and Groups

All mice were sacrificed by deep sodium pentobarbital anesthesia (100 mg/kg) to obtain the tissue. The following sets of experiments were performed:

Experiment 1: The establishment of mouse model with hippocampal *SLC1A1*/EAAT3 knockdown.

Recombinant adeno-associated viral (rAAV) vectors were used to construct the hippocampal *SLC1A1*/EAAT3 knockdown mouse model by hippocampal microinjection. After receiving bilateral hippocampal microinjection of RNA interference vector (RNAi), thirty-six adult mice were randomly divided into six groups (*n* = 6 for each): CON (immediately after microinjection, day 0), D1, D7, D14, D21, and D28. Hippocampal tissues were obtained for RT-qPCR and Western blot analysis.

Subsequently, twenty other adult mice were randomly separated into the NC group and RNAi group based on receiving rAAV-shRNA-NC (NC) or rAAV-shRNA-mSLC1A1 (RNAi) (*n* = 10 for each), respectively. Sixteen days after hippocampal microinjection, the mice were subjected to Morris water maze (MWM) training trails for 5 consecutive days. Twenty-one days after microinjection in the hippocampus, all mice received the MWM probe test for assessing reference memory, and their spontaneous activities were observed in an open field test (OFT) 2 h before the probe test. Then, five mice in each group were randomly selected to perform brain immunofluorescence after OFT.

Experiment 2: Effect of LPS on hippocampal *SLC1A1*/EAAT3 knockdown mice and its possible mechanism.

Adult mice were randomly assigned to four groups: NC + ACSF (NC group), NC + LPS (LPS group), RNAi + ACSF (RNAi group), and RNAi + LPS group based on whether they had to receive rAAV-shRNA-NC (NC), rAAV-shRNA-mSLC1A1 (RNAi), ACSF or LPS (*n* = 12 for each), respectively. Sixteen days after hippocampal microinjection, all mice were given MWM training over the next 5 days. Twenty-one days after microinjection in the hippocampus, all mice received intracerebroventricular microinjection of ACSF or LPS. Furthermore, 24 h later, the mice were conducted to take the MWM probe test. Then, six mice per group were sacrificed and the expression changes of EAAT3 and GluA1 in the hippocampus of mice were detected by Western blot. As for the other mice, their brain tissues were dissected so as for the researchers to observe the morphology changes of pyramidal neurons in the hippocampus by Golgi–Cox staining.

Experiment 3: Effect of riluzole on LPS-induced cognitive impairment in the old mice.

Old mice were randomly divided into four groups, according to whether they had to receive DMSO, riluzole, ACSF, or LPS treatment (*n* = 8 in each group). These groups included DMSO + ACSF (DMSO group), Riluzole + ACSF (Riluzole group), DMSO + LPS (LPS group), and Riluzole + LPS groups. All mice received an intraperitoneal injection of DMSO or riluzole for 3 consecutive days. For the LPS group and Riluzole + LPS group, mice were microinjected LPS into the lateral ventricle 2 h after the last intraperitoneal injection, and equal volumes of ACSF were microinjected in the NC group and Riluzole group. The probe test for reference memory was conducted 1 day after LPS administration, and the hippocampus was obtained for Western blot analysis after the test.

### 2.4. Experimental Methods

Establishment of mouse model with hippocampal *SLC1A1*/EAAT3 knockdown by rAAV-mediated shRNA.

Four different potential shRNA sequences (shRNA-mSLC1A1-1–4) targeting mSLC1A1 and the negative control shRNA (shRNA-NC) were designed and synthesized to construct rAAV vectors, respectively, named rAAV-shRNA-SLC1A1-1–4 and rAAV-shRNA-NC (Gemma Gene Company, Suzhou, China). In order to identify the most effective shRNA sequence to knockdown *SLC1A1*/EAAT3 in the hippocampus, 4 different rAAV vectors were screened to infect the HT-22 cell line. It was observed that rAAV-shRNA-SLC1A1-2 showed the lowest EAAT3 expression by RT/PCR and Western blot (Appendix A).

The mice were anesthetized with pentobarbital sodium (70 mg/kg) and placed on brain stereotaxic apparatus for the process of stereotactic injection of rAAV vector into the bilateral hippocampus, (RWD Life Science, Shenzhen, China). The stereotactic coordinates were 2.1 mm posterior, ±1.7 mm lateral, and 2.0 mm ventral from bregma. Two holes were drilled into the skull for injection with the speed of 50 nL/min, and then the needle was kept in place for an additional 15 min before it was slowly withdrawn. According to the grouping, rAAV-RNAi was injected into the bilateral hippocampus at 1 μL per side (1 × 10^13^ TU/mL) to the RNAi group and RNAi + LPS group, while an equal volume of negative control rAAV-NC was treated to the NC group and NC + LPS group.

Establishment of LPS-induced cognitive impairment model.

In accordance with the previously described LPS-induced cognitive impairment mouse model [18], LPS was administered via the intracerebroventricular (i.c.v.) route. The injection was performed through drilled holes in the skull, into the paracele, using the following stereotactic coordinates (in mm): 0.5 posterior, ±1.0 lateral, and 2.0 ventral from bregma. Every mouse received 2 μL LPS microinjection and the injection speed was set at 0.0667 μL/min. After injection, the needle was left in place for 1 more minute.

Open field test.

In order to evaluate the effects of hippocampal injection of rAAV-RNAi on spontaneous activity in mice, the open field test was carried out 21 days after bilateral hippocampal rAAV-RNAi injection. Mice were placed in the corner of an opaque plastic box (50 cm × 50 cm × 30 cm), whose base was equally divided into 16 parts (4 × 4). A camera was set up immediately above the box to record all the activities of the mice. The considered parameters, such as total moving distance, moving speed, and grid crossing times, were recorded for 5 min and then were analyzed by the ANY-MAZE system. The open field was cleaned with 5% ethyl alcohol and allowed to dry between tests.

MWM test.

The MWM test, a hippocampal-dependent test of spatial learning and memory for rodents, was performed as previously described [22] with minor modifications. The water maze was a stainless-steel circular pool (diameter 125 cm, height 50 cm) with a white inner wall, filled with opaque water containing skimmed milk powder at 22 ± 1.0 °C (water depth 25 cm).

The whereabouts of mice could be recorded by the automatic visual tracking cameras on the ceiling above the pool, which was divided into four equal quadrants—I, II, III, and IV. An escape platform (diameter 10 cm) was fixed in the first quadrant (target quadrant) and submerged 1 cm below the water surface. Spatial learning was evaluated through a 5-day repetitive trial. Mice were randomly released into the pool, facing the wall, and trained to find the platform within 60 s. When a mouse failed to find the platform, it was guided to it. All mice were given four trials (once per quadrant; swim-start position randomized) each day and were allowed to stay on the platform for 10 s. After each daily session, mice were dried under a heater and brought back to the home cage.

After the final acquisition trial, the animals underwent LPS microinjection after 24 h. Then a probe trial was conducted by removing the hidden platform to assess spatial reference memory 24 h after the microinjection. Total swimming distances, average speed, platform-site crossings, the times of entering the target quadrant (original platform quadrant), and time in the target quadrant were recorded. Each mouse was placed in the pool for 60 s at a time, and the starting point of entry was the third quadrant (opposite to the first quadrant).

Real-time reverse transcription polymerase chain reaction (RT-qPCR).

The total RNA samples from mouse hippocampi were extracted using Trizol Reagent (Invitrogen, Waltham, MA, USA, cat no. 15596026) following the Trizol user guide. The operation of cDNA reverse transcription was referred the PrimeScript RT reagent Kit with gDNA Eraser cat no. RR047A protocol (TaKaRa, Kusatsu City, Japan). The operation of RT-qPCR reaction was referred the TB Green Premix Ex Taq II cat no. RR820A protocol (TaKaRa). Primers used for amplifying SLC1A1 were: 5′-AAGAACCCTTTCCGCTTTG-3′ (sense) and 5′-TTGCCGAACTGGACGAGA-3′ (antisense); GAPDH primers were: 5′-CCTTCATTGACCTCAACTACATGG-3′(sense) and 5′-CTCGCTCCTGGAAGATGGTG-3′ (antisense).

Western blot.

The total protein and membrane protein were extracted from hippocampal tissue of the mice by a whole protein extraction kit (KGP250, KeyGEN, Nanjing, China) and a membrane protein and cytoplasmic protein extraction kit (KGP350, KeyGEN, Nanjing, China), respectively. The protein concentrations were determined by BCA protein assay (KeyGEN, Nanjing, China). Equal amounts (30 μg) of proteins were separated by SDS-PAGE and transferred to a polyvinylidene fluoride membrane (Millipore, Billerica, MA, USA). The membrane was blocked with 5% skim milk for 2 h at room temperature, then incubated overnight at 4 °C with the following primary antibody: rabbit anti-EAAT3 (1:1000; Cell Signaling Technology, Danvers, MA, USA), rabbit anti-AMPA Receptor 1 (GluA1) (1:1000; Cell Signaling Technology, Danvers, MA, USA), rabbit anti-Phospho-GluA1-Ser845 (1:1000; Cell Signaling Technology, Danvers, MA, USA), rabbit anti-GAPDH (1:1000, Abcam, Cambridge, UK), and rabbit anti-β-actin antibody (1:1000, Cell Signaling Technology, Danvers, MA, USA). Subsequently, the membrane was washed three times (5 min each) in Tris-buffered saline-Tween 20 (TBST) buffer, incubated in goat anti-rabbit horseradish peroxidase (HRP)-conjugated secondary antibody (1:2000 in TBST buffer) at room temperature for 1 h, and washed again. The optical densities of each protein band were measured with Image Lab software (Bio-Rad, Hercules, CA, USA). Each experiment was repeated at least four times. Relative expression levels of proteins were normalized to β-actin.

Immunofluorescence staining.

The brain tissue of mice was first fixed with 4% paraformaldehyde for 2 days and then embedded in paraffin. Coronal 3 μm sections were prepared and stained with fluorescence immunohistochemistry. In order to repair the antigen, the paraffin sections were first dewaxed and placed in the EDTA buffer (pH 8.0). Thereafter the sections were washed in 0.01% Triton X-100 in phosphate-buffered saline (PBS-T) and blocked with 3% hydrogen peroxide for 15 min at room temperature. The samples obtained were incubated overnight at 4 °C in the appropriate primary antibody, anti-EAAT3 (1:400; Cell Signaling Technology, Danvers, MA, USA). Next, the sections were incubated with the appropriate fluorescent secondary antibody, anti-rabbit IgG (1:400; ZF-0513, Beijing, China), for 30 min at 37 °C. After the secondary antibodies were washed out, the sections were incubated with 4’,6-diamidino-2-phenylindole (DAPI) for nuclear staining. Immunofluorescence was captured with a scanning con-focal microscope.

Golgi–Cox staining.

The Golgi–Cox method is established as one of the most effective techniques for studying the morphology of neuronal dendrites and dendritic spines [23]. The brains of mice were quickly removed and rinsed with double distilled water and were stained with the FD Rapid Golgi Stain kit (FD Neuro Technologies, Ellicott City, MD, USA) following the manufacturer’s instructions. Coronal slices (100 µm thickness) were obtained by using a cryostat (Leica, Wetzlar, Germany), and were then placed on a gelatin-coated microscope slide, stained, and dehydrated. Images were taken by an Eclipse Ci-L microscope (Nikon, Tokyo, Japan) and Image-Pro Plus 6.0 software. Sholl analyses were made with the use of the ImageJ 1.51K Sholl plugin. Spine density was estimated in the number of spines per 10 µm of dendrite length. The number of dendrites was estimated by counting the dendritic intersections, with multiple circular regions of interest centered on the cell soma, with a spacing of 10 μm.

### 2.5. Statistical Analysis

All data collected were analyzed by an observer who was not involved in the experimental protocol. Statistical comparisons between and within groups were made by two-way ANOVA, followed by a Tukey’ test when necessary. For acquisition training, data were analyzed using two-way ANOVA (treatment × trial time) with repeated measures (trial days) followed by the Bonferroni post hoc test. The results were expressed as mean ± standard error of mean (SEM); *p* values < 0.05 were considered as statistically significant.

## 3. Results

### 3.1. Expression of EAAT3 in the Hippocampus of Adult Mice Significantly Decreased 21 Days after rAAV-RNAi Microinjection

The hippocampus tissues were collected at different time points (0, 1, 7, 14, 21, and 28 days after hippocampal microinjection of rAAV-RNAi) to verify the effect of RNAi mediated-knockdown of *SLC1A1/*EAAT3. RT-qPCR and Western blot analysis were used to detect the expression of hippocampal EAAT3 mRNA and the hippocampal EAAT3 protein. The findings showed that the expression of hippocampal EAAT3 mRNA significantly decreased on day 14 (* *p* < 0.05), day 21 (* *p* < 0.05), and day 28 (* *p* < 0.05) compared to the CON group (Figure 1A). Moreover, the expression levels of hippocampal EAAT3 proteins significantly decreased on day 7 (*** *p <* 0.001) and remained downregulated on day 14 (** *p <* 0.01), day 21 (*** *p <* 0.001), and day 28 (*** *p <* 0.001) compared to the CON group (Figure 1B). The immunofluorescence results also showed that the expression of EAAT3 in the hippocampal CA area of RNAi group was significantly lower than NC group on the 21th day after hippocampal microinjection (Figure 1C).

### 3.2. SLC1A1/EAAT3 Knockdown in the Hippocampus Had No Significant Effect on the Learning and Memory of Adult Mice

The MWM test was conducted to examine the effects of hippocampal EAAT3 knockdown on the cognitive function of adult mice. In the MWM test, all mice in the NC group and rAAV-RNAi group learned the original platform location on the fourth day during the acquisition phase. The escape latency in both groups shortened as the training times increased (* *p* < 0.05), but no difference was observed between the two groups on the same day (Figure 2A). In the probe test, there was no significant difference between the two groups on the times of entering the target quadrant (Figure 2C) and the platform-site crossings (Figure 2D) (*p* > 0.05).

### 3.3. SLC1A1/EAAT3 Knockdown in the Hippocampus Significantly Increased the Motility of Adult Mice

The open field test was used to detect the autonomic activity of adult mice 21 days after hippocampal microinjection of rAAV-RNAi/NC. Results obtained showed that the total moving distances (** *p* < 0.01), moving speed (* *p* < 0.05), and times of grid crossing (* *p* < 0.05) in RNAi group significantly increased compared to those in the NC group (Figure 3).

### 3.4. Hippocampal SLC1A1/EAAT3 Knockdown Significantly Aggravated LPS-Induced Learning and Memory Deficit in Adult Mice

Similarly, the MWM test was performed to investigate the effects of hippocampal EAAT3 knockdown on the LPS-induced learning and memory deficit model. All adult mice learned the original platform place on the fourth day during the acquisition phase, and the escape latency of each group also shortened with the increase of training times (** *p* < 0.01, * *p* < 0.05). However, no difference was observed between the groups on the same day (Figure 4A). During the probe test, it was observed that the times of entering the target quadrant of mice in the RNAi group and LPS group were significantly reduced compared with those in the NC group (* *p* < 0.05), and those in the RNAi + LPS group were lower than those in the LPS group (^#^ *p* < 0.05) (Figure 4B). Similarly, the platform-site crossings of mice in RNAi group and LPS group were significantly reduced compared with those in the NC group (* *p* < 0.05), and the platform-site crossings of mice in the RNAi + LPS group were significantly lower than those in the LPS group (^#^ *p* < 0.05) (Figure 4C).

### 3.5. LPS Significantly Decreased Plasma Membrane Protein Level of EAAT3 in Adult Mice

To determine the effects of LPS on EAAT3 protein expression, Western blot analyses were performed to detect both EAAT3 total protein levels and membrane protein levels in the hippocampus. As shown in Figure 5A, the total protein level of EAAT3 in the RNAi group was significantly lower than that in the NC group (* *p* < 0.05), and the total protein level of EAAT3 in the RNAi + LPS group was also significantly lower than that of the LPS group (^#^ *p* < 0.05). However, no significant difference regarding EAAT3 total protein level was discovered between the RNAi group and RNAi + LPS group.

As shown in Figure 5B, plasma membrane protein level of EAAT3 in the LPS group was significantly lower than that of the NC group (** *p* < 0.01), while plasma membrane protein level of EAAT3 in the RNAi + LPS group was significantly lower than the RNAi group (^&&^ *p* < 0.01) and it further decreased compared to that of the LPS group (^#^ *p* < 0.05).

### 3.6. Expression and Phosphorylation of GluA1 Protein Were Inhibited by LPS in the Hippocampus of EAAT3 Knockdown Mice

The effect of EAAT3 knockdown and LPS microinjection on GluA1 expression was also examined in the study. As shown in Figure 6A, the total protein expression level of GluA1 in the RNAi group significantly decreased compared to that in the NC group (* *p* < 0.05), and the total protein expression level of GluA1in the RNAi + LPS group significantly decreased compared to that in the LPS group (^#^ *p* < 0.05).

As shown in Figure 6B, the phosphorylation level of GluA1 protein at the serine 845 site (p-GluA1-Ser-845) in the LPS group was significantly lower than that in the NC group (* *p* < 0.05); in addition, the p-GluA1-Ser-845 level in the RNAi + LPS group was significantly lower than that in the RNA group (^&^ *p* < 0.05) and further decreased compared to that in the LPS group (^#^ *p* < 0.05).

### 3.7. LPS Aggravated the Decreases of Dendritic Complexity in the Hippocampus of EAAT3 Knockdown Mice

To investigate the effect of EAAT3 knockdown and LPS microinjection on dendritic complexity, spine density and dendritic branch numbers of the hippocampus were measured one day after LPS intracerebroventricular injection using Golgi–Cox staining. As shown in Figure 7A, density of dendritic spines markedly decreased in the LPS group (*** *p* < 0.001) and RNAi group (* *p* < 0.05), compared with those in the NC group. Moreover, density of dendritic spines in the RNAi + LPS group were significantly lower than the RNAi group (^&&&^ *p* < 0.001) and further decreased compared to the LPS group (^#^ *p* < 0.05).

Next, as shown in Figure 7B, the intersections of the dendritic branches and the ten concentric circles were counted using Sholl analysis to estimate the number of dendrites. It was discovered that the intersections in the LPS group significantly decreased compared to the NC group (* *p* < 0.05). Moreover, the intersections in the RNAi + LPS group significantly decreased compared to the RNAi group (^&^ *p* < 0.05) and further decreased compared to the LPS group (^#^ *p* < 0.05).

### 3.8. Riluzole Improved Learning and Memory Ability and Ameliorated LPS-Induced Cognitive Impairment in Old Mice

Finally, it was examined whether riluzole could improve the cognitive deficits induced by LPS in elderly mice. During the training phase of the MWM test, all old mice were able to learn the original platform location on the fourth day, and the escape latency in all groups shortened as the training times increased compared with the first day (* *p* < 0.05); yet, no difference was observed between each group on the same day (Figure 8A). During the probe test, it was observed that, compared with the DMSO group, the time in the target quadrant of mice in the Riluzole group significantly increased (* *p* < 0.05), whereas the time in the target quadrant of mice in the LPS group significantly decreased (* *p* < 0.05). In addition, the time in the target quadrant of mice in the Riluzole + LPS group significantly increased compared with the LPS group (^&^ *p* < 0.05) (Figure 8B). Similarly, compared with the DMSO group, the platform-site crossings of mice in the Riluzole group significantly increased (* *p* < 0.05), whereas the platform-site crossings of mice in the LPS group significantly decreased (* *p* < 0.05). Meanwhile, the platform-site crossings of mice in the Riluzole + LPS group were significantly more than the LPS group (^&^ *p* < 0.05) (Figure 8C).

### 3.9. Riluzole Significantly Increased the Expression of EAAT3 Membrane Protein in the Hippocampus of Old Mice

To explore the potential mechanism of riluzole treatment improving learning and memory in old mice, the effect of riluzole on the expression levels of EAAT3 in the hippocampus was investigated using Western blot. For the expression of EAAT3 total protein, there was no significant difference between each group (Figure 9A). However, the expression of EAAT3 plasma membrane protein in the Riluzole group significantly increased compared with that in the DMSO group (* *p* < 0.05). Moreover, the expression of EAAT3 plasma membrane protein in the LPS group significantly decreased compared with that in the DMSO group (* *p* < 0.05). Furthermore, the expression of EAAT3 plasma membrane protein in the Riluzole + LPS group was significantly lower than in the Riluzole group (^#^ *p* < 0.05) (Figure 9B).

## 4. Discussion

Postoperative cognitive dysfunction (POCD) is commonly observed in perioperative care following surgery and general anesthesia in elderly individuals, yet its underlying mechanisms remain largely unknown. So far, no preventive or interventional agents have been established for this condition. Previous studies suggested that extracellular glutamate in the brain increased with age and its dysregulation was associated with impaired learning and memory [24,25,26].

EAAT3 is one of the family of Na^+^-dependent excitatory amino acid transporters that regulate the homeostasis of extracellular glutamate in the CNS, which is encoded by the *SLC1A1* gene and is enriched in the hippocampus neurons [6,7]. Studies have shown that EAAT3 may be downregulated under stress and hyperoxia conditions, leading to glutamate excitotoxicity and neurological abnormality [27]. Importantly, EAAT3 has been discovered to play a critical role in learning and memory [28], and previous study found that lack of EAAT3 led to impaired cognition after isoflurane exposure in EAAT3(–/–) mice [29]. Furthermore, our previous study has shown that the expressions of both EAAT3 total protein and membrane protein in the hippocampus of old mice significantly decreased compared to those of adult mice (Appendix A). Combining this with the higher incidence of POCD in the elderly, it can be inferred that the dysfunction and degradation of EAAT3 expression increased susceptibility to POCD in the elderly. However, the underlying mechanism is still elusive.

Previous studies have used the EAAT3 gene knockout animal model to explore the function of EAAT3 in the CNS [30,31,32]. Considering that animal models with reduced gene expression, rather than deletion, are of more clinical relevance since they better reflect the impact of human polymorphisms on protein levels and rarely complete loss of expression [33]. Therefore, in this study, rAAV-mediated shRNA was constructed to knockdown hippocampal *SLC1A1/*EAAT3 expression via hippocampal microinjection, achieving the effect of local interference and minimizing the impact on the rest of the brain and the whole body [34]. The results of RT-qPCR and Western blot confirmed that bilateral hippocampal injection of RNAi could significantly inhibit hippocampal EAAT3 expression and achieve a stable knockdown effect from 21 to 28 days after RNAi injection. This is consistent with the time point selected by other injection studies of AAV vectors [35,36]. Meanwhile, the specific expression of GFP (green fluorescent protein) was observed in the entire CA region 21 days after the hippocampal injection, indicating that the rAAV microinjection was reliable (Appendix A). In behavioral experiments, there was no significant difference between NC and RNAi group in the times of entering target quadrant and the platform-site crossings during the probe test, which indicated that EAAT3 knockdown in the hippocampus had no significant effect on cognitive function of adult mice under normal conditions. Furthermore, it was discovered that autonomic activity (including total moving distance, moving speed, and times of grid crossing) in the RNAi group was significantly higher than that in the NC group, which might be caused by the increased glutamate between synaptic space induced by the interference with EAAT3 expression. Glutamate acts as the major excitatory neurotransmitter and mediates fast neurotransmission in neuronal networks [37]. Interference with EAAT3 expression can disrupt the absorption of extracellular glutamate and lead to neuronal hyperexcitation and even excitotoxicity injury [38]. However, the role of EAAT3 in POCD has been rarely explored.

The method of intracerebroventricular administration of LPS to construct the POCD model has been applied by our team in many researches and has achieved substantial results [19,20,39]. Therefore, in the present study, the same POCD model of mice was constructed based on our previously described approach. The study revealed that LPS led to significant learning and memory deficits and interference with EAAT3 expression by rAAV-RNAi significantly aggravated the learning and memory impairment induced by LPS. These results indicated that EAAT3 dysfunction increased susceptibility to LPS-mediated learning and memory deficits.

Under basal conditions, EAAT3 is primarily sequestered in the intracellular compartment, with about 20% of the transporter localized at the cell surface [40], which is a key determinant of its buffering efficiency [41] and accounts for 40% of glutamate uptake in the hippocampus [5]. In this study, it was found that LPS treatment did not significantly decrease the EAAT3 total protein level, either in the NC group or in the RNAi group, but significantly decreased the EAAT3 plasma membrane protein level in both the NC group and the RNAi group, thus indicating that LPS mainly disrupted the expression of EAAT3 plasma membrane protein. Interestingly, it was discovered that in the ACSF group, rAAV-RNAi only significantly decreased EAAT3 total protein level but did not significantly decrease EAAT3 membrane protein level. However, in the LPS treatment group, rAAV-RNAi significantly decreased both EAAT3 total protein level and EAAT3 membrane protein level, which further corroborated that LPS primarily disrupted EAAT3 protein trafficking to the plasma membrane. So far, no study has reported this finding. Yet, the expression of *SLC1A1*, encoding EAAT3, has been found to be impaired in neuroinflammation-related diseases, such as multiple sclerosis, schizophrenia, hypoxia/ischemia, and epilepsy [42]. This suggests that neuroinflammation induced by LPS may lead to the disruption of trafficking of EAAT3 protein to plasma membrane. Nevertheless, the underlying mechanisms need to be further explored.

Trafficking of AMPAR to the plasma membrane in the neurons is believed to be a fundamental cellular process for learning and memory [43,44,45], mediated via the phosphorylation of GluA1 (an AMPAR subunit) at serine 845 (P-GluA1-s845) and then promoting AMPAR trafficking to the plasma membrane [44,45,46]. In this study, it was discovered that LPS treatment did not significantly decrease the total GluA1 protein level in the NC group or in the RNAi group. However, it significantly decreased the P-GluA1-s845 level, which indicated that LPS primarily interfered with the P-GluA1-s845 level. In addition, it was discovered that interference with *SLC1A1/*EAAT3 expression by rAAV-RNAi significantly decreased both the GluA1 protein level and the P-GluA1-s845 level, indicating that EAAT3 could affect the expression of GluA1 and P-GluA1-s845. These findings were consistent with our previous study that EAAT3 could regulate GluA1 trafficking to the plasma membrane [47]. These results indicated that LPS might disrupt the GluA1 phosphorylation (s845) and the trafficking of AMPAR to the plasma membrane by interfering with EAAT3 plasma membrane expression, and then impair the cognitive function of adult mice. Therefore, EAAT3 may become a hopeful therapeutic target for the treatment of POCD.

Dendritic spines, defined as small protrusions arising from dendrites, are used for communication by excitatory glutamatergic synapses of the CNS. They are remarkably dynamic structures that may undergo adaptive changes following different stimuli [48], and their plasticity is thought to be crucial to memory formation [49]. In this study, it was observed that LPS treatment significantly decreased dendritic complexity of the hippocampal neurons in NC and RNAi group, including the spine density and dendritic branches numbers. Many studies have shown that LPS could significantly reduce the dendritic spine density in hippocampal neurons [39,50] by activating microglia to hyperactively secret pro-inflammatory cytokines [51]. In addition, glutamate excitotoxicity, which could be induced by the interference of EAAT3 expression, may be a mechanism that causes secondary damage to neurons, accompanied by loss of dendritic spines and changes in synaptic activity [52,53]. Moreover, it was also observed in this study that interference with EAAT3 expression significantly decreased the dendritic spine density. Data collected further indicated that EAAT3 knockdown aggravated LPS-induced impairment of dendritic complexity. Thus, it can be inferred that EAAT3 dysfunction may be involved in increased susceptibility to LPS-mediated learning and memory deficits.

To further verify the important role of EAAT3 in POCD, we used riluzole pretreatment to explore whether riluzole could ameliorate POCD in old mice via regulating EAAT3 expression. Our study discovered that in old mice, riluzole pretreatment significantly promoted their learning and memory function and increased the EAAT3 membrane protein level, but not EAAT3 total protein level. This demonstrated that riluzole could improve the cognitive function of old mice by promoting EAAT3 membrane protein level. In LPS-treated old mice, riluzole pretreatment significantly alleviated LPS-induced learning and memory deficits, which may be related to its upregulation effect of EAAT3 membrane protein level. However, there was no significant difference in the trend of upregulation, which may be due to the serious disruption by LPS of EAAT3 membrane protein in old mice, and riluzole may have other protective mechanisms to improve the cognitive impairment of LPS-treated old mice. Therefore, it can be inferred that changes of EAAT3 function play a significant role in the pathogenesis and the treatment of POCD, and which may become a promising target for POCD treatment.

## 5. Conclusions

This study revealed that the decrease of EAAT3 expression and function was associated with advanced age, and the decrease of EAAT3 expression aggravated the cognitive impairment induced by LPS. Inflammation induced by LPS significantly decreased the plasma membrane expression of EAAT3 in the hippocampus, which in turn inhibited the phosphorylation of Glua1—the mechanism of AMPAR transport to the plasma membrane, and reduced synaptic density in hippocampal neurons. It implied that EAAT3 plasma membrane protein level in the hippocampus was closely linked with learning and memory, and this may be the reason why EAAT3 knockdown increases the susceptibility of the cognitive impairment induced by LPS.

## Figures and Tables

**Figure 1 membranes-12-00317-f001:**
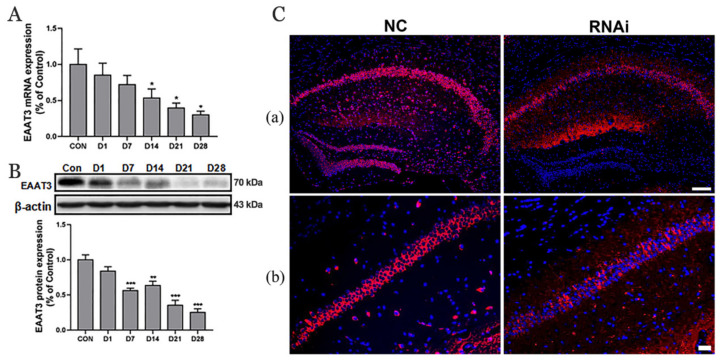
Expression of EAAT3 in the hippocampus of adult mice decreased after rAAV-RNAi microinjection. (**A**,**B**) Relative mRNA (**A**) and protein (**B**) expression of EAAT3 in the hippocampus after receiving virus injection. (**C**) Representative images of hippocampal EAAT3 expression 21 days after viral injection. Expression of EAAT3 protein in the whole hippocampus (**C**(**a**)) and in the hippocampal CA regions (**C**(**b**)). Data are expressed as mean ± SEM (*n* = 6). The scale bar in Figure (**C**(**b**)) is 100 μm and in Figure (**C**(**b**)) is 20 μm. * *p* < 0.05, ** *p* < 0.01, and *** *p* < 0.001 vs. control. CON, control (immediately after microinjection of the hippocampus); D1, 7, 14, 21, 28 (1 day, 7 days, 14 days, 21 days, 28 days after microinjection).

**Figure 2 membranes-12-00317-f002:**
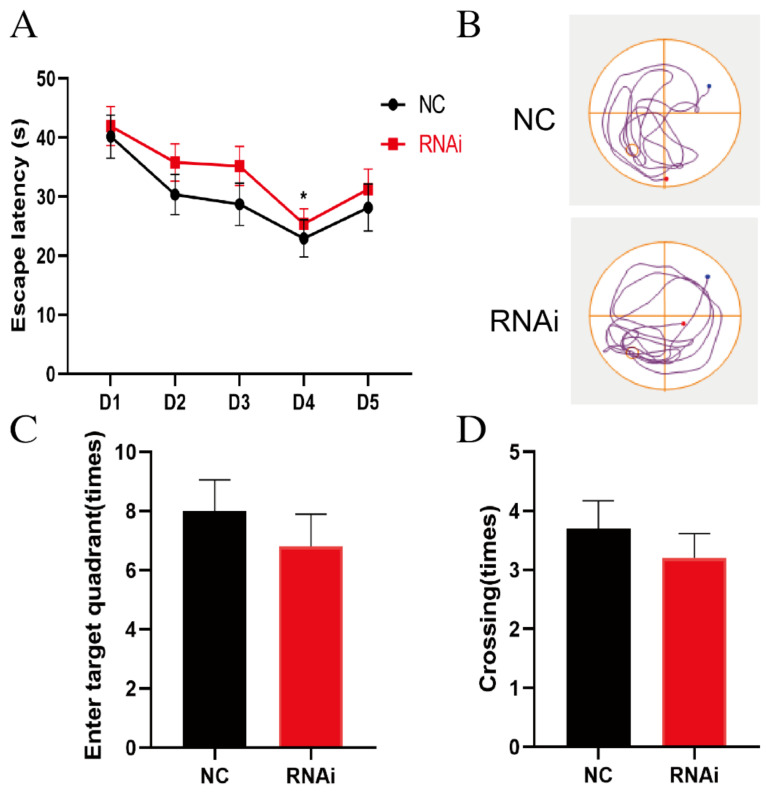
EAAT3 knockdown in the hippocampus had no significant effect on the learning and memory of adult mice. (**A**) Both NC and RNAi groups of mice learned to locate the hidden platform by using the surrounding cues after 5-day training. (**B**) Representative trajectories of adult mice in the probe trial of the MWM test, the blue and red dots respectively represent the beginning and end of the trajectory, and the circle represents the previous location of platform. (**C**,**D**) During the probe trial, the hidden platform was removed and the times of entering target quadrant (**C**) as well as the platform-site crossings (**D**) were analyzed. Data are expressed as mean ± SEM (*n* = 10). * *p* < 0.05 vs. day 1.

**Figure 3 membranes-12-00317-f003:**
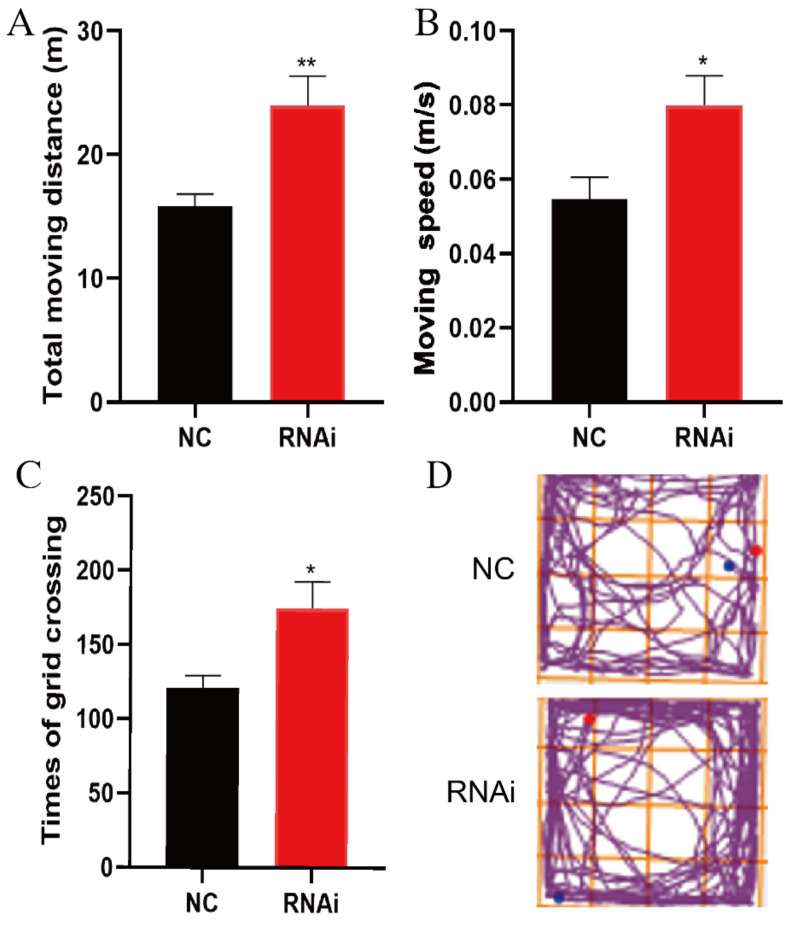
EAAT3 knockdown in the hippocampus increased motility of adult mice. (**A**–**D**) Total moving distance (**A**) and moving speed (**B**) are expressed as the mean ± SE; times of grid crossing (**C**) are expressed as mean ± SEM (*n* = 10); (**D**) Representative moving traces in the open arena, the blue and red dots respectively represent the beginning and end of the trajectory. * *p* < 0.05, ** *p* < 0.01 vs. NC group.

**Figure 4 membranes-12-00317-f004:**
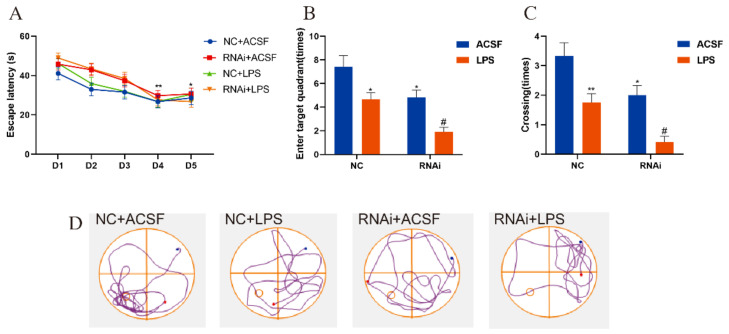
EAAT3 knockdown in the hippocampus aggravated LPS-induced learning and memory deficits in adult mice. (**A**) All experimental groups of mice learned to locate the hidden platform by using the surrounding cues after 5-day training. (**B**,**C**) During the probe trial, the hidden platform was removed and the times of entering target quadrant (**B**) as well as the platform-site crossings (**C**) were analyzed. (**D**) Representative trajectories of adult mice in the probe trial of MWM test, the blue and red dots respectively represent the beginning and end of the trajectory, and the circle represents the previous location of the platform. Data are expressed as mean ± SEM (*n* = 12). * *p* < 0.05 vs. NC group, ** *p* < 0.01 vs. NC group, ^#^ *p* < 0.05 vs. LPS group.

**Figure 5 membranes-12-00317-f005:**
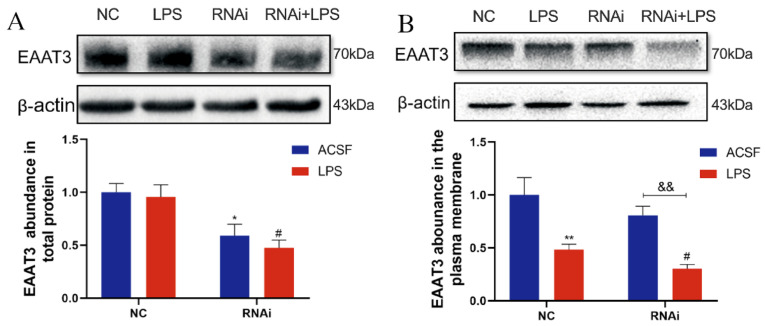
Effects of lateral ventricle microinjection of LPS on hippocampal EAAT3 expression in adult mice. (**A**,**B**) Protein bands on the gel and their relative intensities. The expression levels of total protein (**A**) and plasma membrane protein (**B**) of EAAT3 in the hippocampus of adult mice were normalized to β-actin as an internal control. Data are expressed as mean ± SEM (*n* = 6). * *p* < 0.05, ** *p* < 0.01 vs. NC group, ^#^ *p* < 0.05 vs. LPS group, ^&&^ *p* < 0.01 vs. RNAi group.

**Figure 6 membranes-12-00317-f006:**
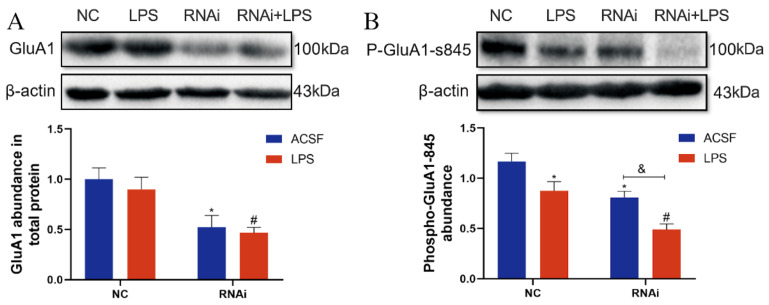
Expression and phosphorylation of GluA1 protein were inhibited by LPS-induced neuroinflammatory in the hippocampus of EAAT3 knockdown mice. (**A**,**B**) Protein bands on the gel and their relative intensities. The expression levels of GluA1 protein (**A**) and Phospho-GluA1-Ser845 in the total protein (**B**) in the hippocampus were normalized to that of β-actin as an internal control. Data are expressed as mean ± SE (*n* = 6). * *p* < 0.05 vs. NC group, ^#^
*p* < 0.05 vs. LPS group, ^&^ *p* < 0.05 vs. RNAi group.

**Figure 7 membranes-12-00317-f007:**
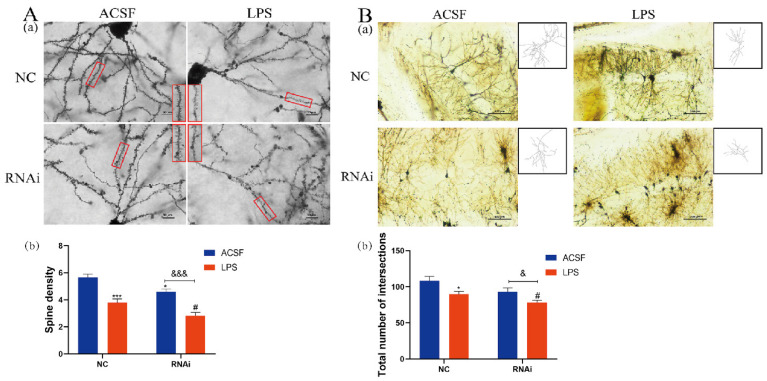
LPS aggravated the decreases of dendritic complexity in the hippocampus of EAAT3 dysfunctional mice. (**A**,**B**) Representative photomicrograph of dendritic spine, magnified view of red boxed areas is shown on the Right side of NC group, RNAi group and the Left side of LPS group, RNAi + LPS group (**Aa**) and dendritic branch (**Ba**) from each group. The spine density (**Ab**) as well as the dendritic branch numbers (**Bb**) of hippocampal neurons were measured by Golgi–Cox staining one day after LPS intracerebroventricular injection. Data are expressed as mean ± SEM (*n* = 6). For the spine density, * *p* < 0.05, *** *p* < 0.001 vs. NC group, ^#^ *p* < 0.05 vs. LPS group, ^&&&^ *p* < 0.001 vs. RNAi group. For the dendritic branch numbers, * *p* < 0.05 vs. NC group, ^#^ *p* < 0.05 vs. LPS group, ^&^ *p* < 0.05 vs. RNAi group. The scale bar in (**A**) is 10 μm and in (**B**) is 100 μm.

**Figure 8 membranes-12-00317-f008:**
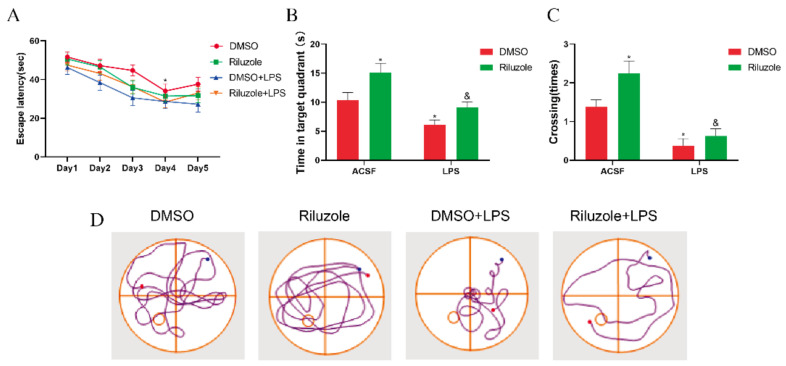
Riluzole improved learning and memory ability and LPS-induced cognitive impairment in old mice. (**A**) All experimental groups of mice learned to locate the hidden platform by using the surrounding cues after 5-day training. (**B**,**C**) During the probe trial, time in the target quadrant (**B**) and platform-site crossings (**C**) were analyzed and expressed as the mean ± SEM (*n* = 8), respectively. (**D**) Representative trajectories from each group in probe trial; the blue and red dots respectively represent the beginning and end of the trajectory, and the circle represents the previous location of the platform. * *p* < 0.05 vs. NC group, ^&^ *p* < 0.05 vs. LPS group.

**Figure 9 membranes-12-00317-f009:**
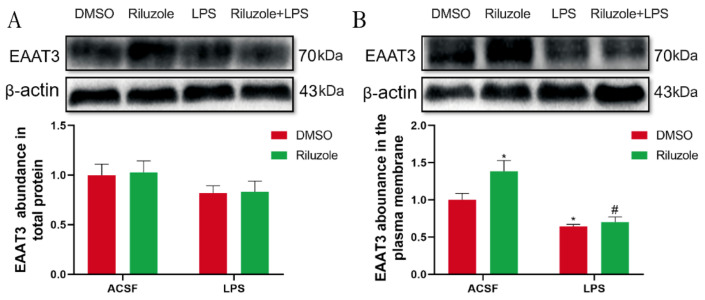
Riluzole increased the expression of EAAT3 membrane protein in the hippocampus of elderly mice. (**A**,**B**) Protein bands on the gel and their relative intensities. The expression levels of total protein (**A**) and plasma membrane protein (**B**) of EAAT3 in the hippocampus of aged mice were normalized to β-actin as an internal control. Data are expressed as mean ± SEM (*n* = 4). * *p* < 0.05 vs. NC group, ^#^ *p* < 0.05 vs. Riluzole group.

## Data Availability

The data presented in this study are available on request from the corresponding author. The data are not publicly available due to privacy or ethical restrictions.

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
