# Peer review of "Dysfunction of EAAT3 Aggravates LPS-Induced Post-Operative Cognitive Dysfunction"

_membranes, 2022, doi:10.3390/membranes12030317_

Round 1

Reviewer 1 Report

Figure 1's statistics are in doubt. How does D14 in 1B not make sense? and the figures statistics do not match with the bands shown in part 1C.

It is seen that there are no references from the last three years in the Introduction and Discussion sections. This situation calls into question the topicality of the article. Please discuss by citing current high impact factor articles.

I think it's a minor revision.

Author Response

Response to Reviewer 1 Comments

Point 1: Figure 1's statistics are in doubt. How does D14 in 1B not make sense? and the figures statistics do not match with the bands shown in part 1C.

Response 1: Thanks for your valuable and scientific comments. We must admit that in the original manuscript, the statistics of Figure 1 are incorrect. The D14 in 1B in fact make sense. We have corrected figure 1 and the statistics have match with the bands shown in part 1C.

Point 2: It is seen that there are no references from the last three years in the Introduction and Discussion sections. This situation calls into question the topicality of the article. Please discuss by citing current high impact factor articles.

Response 2: Thanks for your scientific suggestion. According to your suggestion, we searched PubMed again for high impact factor articles in recent three years, and conducted supplementary discussion in the Introduction and Discussion sections.

Reviewer 2 Report

The authors showed that EAAT3 knockdown aggravated LPS-induced dysfunction of spatial memory in mouse, and biochemical analyses showed that both EAAT3 knockdown and LPS injection decreased GluA1 levels as well as its phosphorylation levels in membrane fraction derived from mouse brain. Finally, they showed that Riluzole, a glutamate transporter activator, rescued not only age-dependent but also ameliorated LPS-induced dysfunction of spatial memory in mice.

The data are intriguing, however, it sounds unreasonable to use LPS induction model as postoperative cognitive dysfunction (POCD) model. Since LPS injection causes neuroinflammation directly and locally, it does not refrect secondarily effects caused by anesthesia or surgery. It would be reasonable to consider that EAAT3 is involved in local neuroinflammation and Rilusole treatment can be effective to ameliorate such direct neuroinflammatiuon.

In Fig.2, the authors shouwed that EAAT3 knockdown did not affect spatial memory in mice. However, Fig.4 shouwed that EAAT3 knockdown clearly impaired memory function by itself. This is a big discrepancy. Which is true?

In Fig.7, the authors examined Golgi-Cox staining to evaluate synaptic preservation. Additional biochemical analyses of pre- and post-synapse marker proteins, such as  synaptophysin and PSD95, would strongly support this finding.

Minor points:

1) Figurements should be rearranged according to its description. For example, in the current manuscript, the authors described Fig.1B prior to Fig.1A.

2) Fig.5, Fig. 6, and Fig, 9 lack information on the immunoblot lane. The authors should indicate which lane is NC, LPS, and so on.

Author Response

Response to Reviewer 2 Comments

Point 1: The data are intriguing, however, it sounds unreasonable to use LPS induction model as postoperative cognitive dysfunction (POCD) model. Since LPS injection causes neuroinflammation directly and locally, it does not refrect secondarily effects caused by anesthesia or surgery. It would be reasonable to consider that EAAT3 is involved in local neuroinflammation and Rilusole treatment can be effective to ameliorate such direct neuroinflammatiuon.

Response 1: Thanks for your kind suggestion. As you said, LPS induction model is not a perfect model of POCD. Although LPS injection causes neuroinflammation directly and locally, it’s still a model of POCD and have been used in many other studies (Below are the relevant references). Therefore, this model should still be able to reflect the disease state of POCD and can be used for related research.

1.Liu Y, Zhang Y, Zheng X, Fang T, Yang X, Luo X, Guo A, Newell KA, Huang XF, Yu Y. Galantamine improves cognition, hippocampal inflammation, and synaptic plasticity impairments induced by lipopolysaccharide in mice. J Neuroinflammation. 2018;15:112.

2.Zhang XY, Cao JB, Zhang LM, Li YF, Mi WD. Deferoxamine attenuates lipopolysaccharide-induced neuroinflammation and memory impairment in mice. J Neuroinflammation. 2015;12:20.

3.Zhao WX, Zhang JH, Cao JB, Wang W, Wang DX, Zhang XY, Yu J, Zhang YY, Zhang YZ, Mi WD. Acetaminophen attenuates lipopolysaccharide-induced cognitive impairment through antioxidant activity. J Neuroinflammation. 2017;14:17.

Point 2: In Fig.2, the authors showed that EAAT3 knockdown did not affect spatial memory in mice. However, Fig.4 showed that EAAT3 knockdown clearly impaired memory function by itself. This is a big discrepancy. Which is true?

Response 2: Thanks very much for your valuable and scientific comment. We are so sorry to make you confused and we would like to explain this. As shown in Fig. 2, the spatial memory of EAAT3 knockdown model was not significantly affected in the MWM probe test twenty-one days after microinjection. However, cognitive function in RNAi+ACSF group was significantly lower than in NC+ACSF group. One reason could account for this difference. The only difference between the experiments in Fig. 2 and Fig. 4 is that the two groups of mice in Fig. 4 underwent another craniotomy to inject ACSF, and this surgical stress may cause a further decline in the cognitive function of the RNAi group mice, which also indirectly demonstrated that EAAT3 knockdown mice were more susceptible to cognitive impairment under surgical or anesthesia stress.

Point 3: In Fig.7, the authors examined Golgi-Cox staining to evaluate synaptic preservation. Additional biochemical analyses of pre- and post-synapse marker proteins, such as  synaptophysin and PSD95, would strongly support this finding.

Response 3: Thanks for your kind suggestion, which was of great importance for our finding. According to your suggestion, the levels of synapse-related proteins concluding SYN-1 and PSD95 will be further detected in our future studies.

Point 4: Figurements should be rearranged according to its description. For example, in the current manuscript, the authors described Fig.1B prior to Fig.1A.

Response 4: Thanks for your reminding. According to your scientific suggestion, we rechecked the arrangement order of the illustrations in the manuscript and corrected them in the corresponding position.

Point 5: Fig.5, Fig. 6, and Fig, 9 lack information on the immunoblot lane. The authors should indicate which lane is NC, LPS, and so on.

Response 5: Thanks for your reminding. According to your scientific suggestion, we marked the name of each group above the corresponding immunoblot lane for understanding.

Reviewer 3 Report

The authors investigated the dysfunction of EAAT3 on learning, memory and hippocampal dendritic morphology in mice. The EAAT3 dysfunction was induced through viral knockdown technique. The EAAT3 knockdown animals were treated with i.c.v. LPS in order to induce the POCD in these mice. Riluzole was used to treat POCD and improve cognitive functions of old animals. It turned out that EAAT3 dysfunction alone did not affect the cognitive functions of the animals significantly - although the escape latency in the MWM increased slightly in EAAT3 knockdown animals. However, EAAT3 deficiency significantly aggravated the POCD in these animals. Riluzole improved these symptoms significantly. Riluzole was also useful in old animals with decreased EAAT3 function. The results contribute to the understanding of the  pathomechanism of POCD and the role of EAAT3 in cognitive processes in the hippocampal formation. There minor spelling mistakes in the manuscript which have to be corrected.

Author Response

Response to Reviewer 3 Comments

Point 1: The authors investigated the dysfunction of EAAT3 on learning, memory and hippocampal dendritic morphology in mice. The EAAT3 dysfunction was induced through viral knockdown technique. The EAAT3 knockdown animals were treated with i.c.v. LPS in order to induce the POCD in these mice. Riluzole was used to treat POCD and improve cognitive functions of old animals. It turned out that EAAT3 dysfunction alone did not affect the cognitive functions of the animals significantly - although the escape latency in the MWM increased slightly in EAAT3 knockdown animals. However, EAAT3 deficiency significantly aggravated the POCD in these animals. Riluzole improved these symptoms significantly. Riluzole was also useful in old animals with decreased EAAT3 function. The results contribute to the understanding of the  pathomechanism of POCD and the role of EAAT3 in cognitive processes in the hippocampal formation. There minor spelling mistakes in the manuscript which have to be corrected.

 Response 1: Thanks for your valuable suggestion. According to your suggestion, we have carefully checked the spelling of the words in the manuscript and corrected the misspelled ones.